# Spatial pattern of COVID-19 deaths and infections in small areas of Brazil

**Everton Emanuel Campos de Lima****[1], Ezra Gayawan[2], Emerson Augusto Baptista[3]\*, Bernardo Lanza Queiroz[4]**

**1** College of Philosophy and Human Sciences (IFCH), University of Campinas (UNICAMP), Campinas, Brazil, **2** Department of Statistics, Federal University of Technology, Akure, Nigeria, **3** Asian Demographic Research Institute (ADRI), Shanghai University, Shanghai, China, **4** Department of Demography, Universidade Federal de Minas Gerais (UFMG), Belo Horizonte, Brazil

\* emersonaug@gmail.com

## Abstract

As of mid-August 2020, Brazil was the country with the second-highest number of cases and deaths by the COVID-19 pandemic, but with large regional and social differences. In this study, using data from the Brazilian Ministry of Health, we analyze the spatial patterns of infection and mortality from Covid-19 across small areas of Brazil. We apply spatial autoregressive Bayesian models and estimate the risks of infection and mortality, taking into account age, sex composition of the population and other variables that describe the health situation of the spatial units. We also perform a decomposition analysis to study how age composition impacts the differences in mortality and infection rates across regions. Our results indicate that death and infections are spatially distributed, forming clusters and hot-spots, especially in the Northern Amazon, Northeast coast and Southeast of the country. The high mortality risk in the Southeast part of the country, where the major cities are located, can be explained by the high proportion of the elderly in the population. In the less developed areas of the North and Northeast, there are high rates of infection among young adults, people of lower socioeconomic status, and people without access to health care, resulting in more deaths.

## Introduction

The COVID-19 pandemic has proved to be a challenge that is capable of impacting public health in all countries of the world with outcomes hitherto unimaginable. However, for several reasons, including political and economic reasons, some countries, like Brazil, have faced additional difficulties in controlling the pandemic. Brazil has become the new epicenter of this disease [1] as a result of the actions of many local authorities who disregarded WHO recommendations and did not implement strong measures such as population isolation, mask wearing, testing and contact tracing [2–4].

Brazil is characterized by large regional and socioeconomic disparities and high levels of inequality in access to health services. Thus, one may expect these features to impact the country's number of infections and deaths from COVID-19 at different levels across regions of the

**Data Availability Statement:** This data is public available at: https://opendatasus.saude.gov.br/dataset/casos-nacionais.

**Funding:** Gayawan contribution was funded by the São Paulo Research Foundation (FAPESP) grant

268 2018/18649-7. Lima and Gayawan thank the University of Campinas Research Fund - 269 FAEPEX for support.

**Competing interests:** The authors have declared that no competing interests exist.

country. Additionally, at the onset of the pandemic, there was a positive correlation between deaths and infections rates in a few cities in the southern part of the country [5], but recently, the disease has also spread through less developed areas of the country [6–9].

Besides health disparities, there are three components that need to be considered when studying the pattern of COVID-19 in Brazil. First, mortality rates are marked by a considerable age gradient, that is, the number of deaths is significantly higher among the elderly and, consequently, the population age structure is important for understanding the spread and the risk of mortality across areas [10–12]. Second, the number of deaths is significantly higher among males [13]. We hypothesized that since women give more attention to issues affecting their health, this could have a positive impact on mortality risk. In addition, women are the majority of informal workers, domestic workers and the unemployed [14], that is, they are also more directly affected by the lockdown imposed in many parts of the country. Finally, as is the case elsewhere, pre-existing health problems or comorbidities seem to increase mortality risk in the population [12,15]. Some studies have shown that mortality is significantly higher in population strata that present any history of cardiovascular diseases, diabetes or obesity [12,16,17]. The combination of these three components–age structure, comorbidities and sex–may also play a role in the final number of deaths.

In this paper, we used publicly available data from the Brazilian Ministry of Health to estimate infection and mortality rates from COVID-19 in small-areas, taking into consideration the population age structure and sex distribution, and the level of income inequality across regions. We accounted for the number of intensive care units (ICUs) and number of physicians available in each municipality in order to consider the possible relationship with cases and deaths from COVID-19. We used a Bayesian hierarchical model for the 558 micro-regions of Brazil, based on data available at the end of July. Estimates at the local level are very important to develop proper public health interventions and evaluate the impact of the pandemic. An important caveat is that we used information on cumulative infections and death counts on a specific date. We did not control for the period in which the pandemic started in each area of the country, but we used the most recent data available to investigate how the pandemic has spread across the country. In addition, in using the cumulative count of cases and deaths in municipalities that had counts, we assumed that other municipalities had zero counts as of the date the first cases were recorded. This explains why, in the case of mortality, we used a zero-inflated Poisson model, since no death was recorded in many of the municipalities as at the time of the study. Notwithstanding, we observed that areas where the pandemic spread later went on to report significant increases and are rapidly catching-up, in terms of cases and mortality, to the areas that began to experience the pandemic first.

## Methods

### Data source and level of analysis

We use public available data from the Ministry of Health of Brazil, DATASUS, available at https://opendatasus.saude.gov.br/dataset/casos-nacionais [18]. Since 2009, there has been a system of surveillance for Serious Acute Respiratory Syndrome (SARS) that tracks influenza, SARS and other respiratory illnesses across the country. The system incorporates recent information on COVID-19 deaths and infections at the municipal level (defined as a single political boundary), the smallest geographical aggregation of the country. In addition, data on intensive care unit (ICUs) and number of physicians available in each municipality in 2019 were extracted from the Instituto Brasileiro de Geografia e Estatística (IBGE), which is also publicly available (https://mapasinterativos.ibge.gov.br/covid/saude/).

Noronha et al [19] show that the number of health care beds in Brazil is about half of what is observed in OECD countries. Moreover, the spatial distribution is very unequal across the country. According to their analysis, the regions with lowest supply are concentrated in Amazonas, Pará, Roraima, Minas Gerais and Ceará. The supply of ICUs is also very unequal across the country, with most of regions in the North and Northeast showing the lowest supply of tertiary care, which is highly concentrated in the state capitals.

Due to instability in the numbers of cases and the size of the exposed population, we aggregated the municipality information into 558 microregions, as defined by the Brazilian National Statistical Office (IBGE). These geographical areas are aggregated according to regional and socioeconomic similarities, which provide a good benchmark to compare spatial distribution of COVID-19 disease across the country.

We collected the information on July 20th, when Brazil had registered 77,226 deaths in this database and a total of 1,795,054 infections. There were 45,013 deaths of males and 32,213 of females–a male/female ratio of 1.39. For infections, there is a different picture, a male/female ratio of 0.89, with 846,744 and 948,310 infections for males and females, respectively.

## Statistical method

We used a Bayesian formulation of a spatial autoregressive model (SAM), suitable for smoothing and stabilizing spatial estimates, to study the spatial pattern of the infections and mortality [20].

The observed count of cases $Y_i$ in area $i$ is modelled using a Poisson distribution with mean $E_i\theta_i$, where $E_i$ is the expected count of cases and $\theta_i$ is the relative risk in area $i$. The relative risk ($\theta_i$) quantifies whether an area has a higher ($\theta_i>1$) or lower ($\theta_i<1$) risk than the average risk in the standard population. In the case of mortality, because of the large number of zeros recorded in most of the study areas, a zero-inflated Poisson model was considered appropriate. If $Y_i \sim Po(E_i\theta_i)$, $i = 1,...n$, then the relative risk ($\theta_i$) can be modelled through a log link function such that

$$log(\theta_i) = \alpha + u_i + v_i + \gamma_i + f(icu) + f(phy) \qquad (1)$$

Where $\alpha$ is the overall risk in the study areas, $(u_i+v_i)$ is an area random effect, such that $u_i$ models the spatial dependency between the relative risks and $v_i$ is an unstructured exchangeable component that models uncorrelated noise, such that $v_i \sim N(0, \sigma_v^2)$, while $\gamma_i$ is a function assumed for the age categories. The functions $f(icu)$ and $f(phy)$ are smooth functions assumed to model the possible nonlinear effects of number of ICUs and number of physicians. The structured spatial component was considered through the conditional autoregressive (CAR) distribution, which smooths the risks of certain neighbourhood structures, considering two areas neighbours if they share a common boundary. The prior is specified as:

$$(u_i|u_{-i}) \sim N\left(\bar{u}_{\delta_i}, \frac{\delta_u^2}{n_{\delta_i}}\right),$$

Where $\bar{u}_{\delta_i} = n_{\delta_i}^{-1}\sum_{j\in\delta_i}u_j$, $\delta_i$ represents the set of neighbours and $n_{\delta_i}$ is the number of neighbours of area $i$. The unstructured component $v_i$ was considered as independent and identically distributed normal with zero mean and variance $\delta_v^2$. The functions $\gamma_i$, $f(icu)$, and $f(phy)$ were modelled through a random walk of order 2 prior, and $\gamma_i$ was also considered as a linear effect and modelled together with the intercept using Gaussian priors with mean 0 and precision 0.001.

The Bayesian inference was determined using the integrated nested Laplace approximation (INLA) [21], as implemented in the R-INLA package [22]. For each of cases and mortality we considered four models for each gender by varying the forms and terms in model 1 and

compared model performance through the deviance information criterion (DIC) and Watanabe information criterion (WAIC), where the model with the lowest value for each of the criterion is considered the best. Results reported are based on the best model.

## Decomposition of mortality and infection risk according to age structure

In addition to the spatial analysis, we performed a decomposition analysis to investigate the possible effects of population age structure on mortality and infections. Thus, we sought to analyze the contribution of this factor on possible regional differences in the risks of death and infection across the regions [23].

We decomposed each difference in mortality and infection risk between two regions based on Kitagawa's [24] formulation:

$$R_2 - R_1 = \sum_{i=1}^{n}(P_{2i} - P_{1i}) \times \frac{(R_{2i} - R_{1i})}{2}) + \sum_{i=1}^{n}(R_{2i} - R_{1i}) \times \frac{(P_{2i} - P_{1i})}{2}) \tag{2}$$

Where, $R_2$ and $R_1$ are the overall risks of an outcome in populations 2 and 1, and $i$ is the category of predictor variable, age. $P_{2i}$ and $P_{1i}$ are the proportion of population 2 and 1 in category i of the predictor, and $R_{2i}$ and $R_{1i}$ are the risks of the outcome in population 2 and 1 in category i of the predictor.

## Results

Fig 1 presents the box plot of the posterior means of the relative risks of morality by age group from all the microregions. Mortality risks from COVID-19 are low in the age group 0–4 years

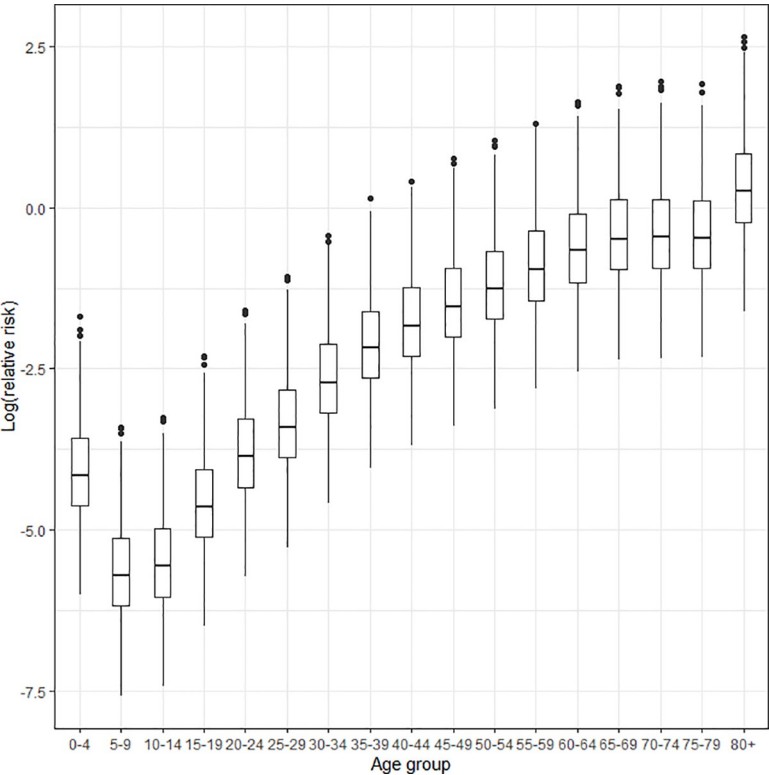

**Fig 1. Log relative mortality risk according to conventional five-year age groups for both sexes in Brazil, 2020.**
Source: Brazilian Ministry of Health 2020.

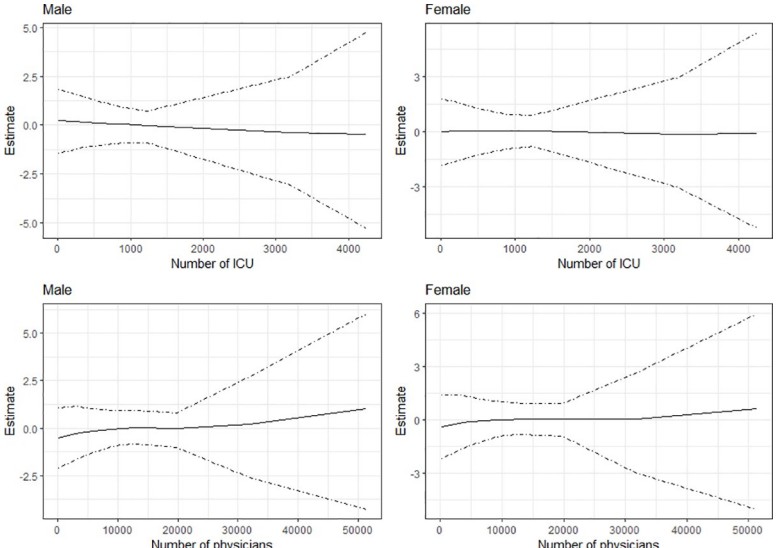

**Fig 2. Estimated effects of number of ICUs and number of physicians on infections from COVID-19 in the municipalities of Brazil.**

but increase steadily with age. Though, on average, the relative risks for all the microregions are below one unit across the age groups, the Fig 1 reveals that from the 40–44 age group on, there are several regions standing as outliers, where the risks are above one, and this is more pronounced for individuals aged 80 years and above. These patterns of risks are observed for males and females, but mortality risks for males are greater than for females across all ages.

Fig 2 presents, for infection from COVID-19, the estimated effects for number of ICUs and number of physicians respectively, while Fig 3 presents the results for mortality. The Figs show the posterior mean estimates (middle solid lines) surrounded by the 95% credible intervals. For all the plots, the credible intervals become wider as the number of ICUs and physicians

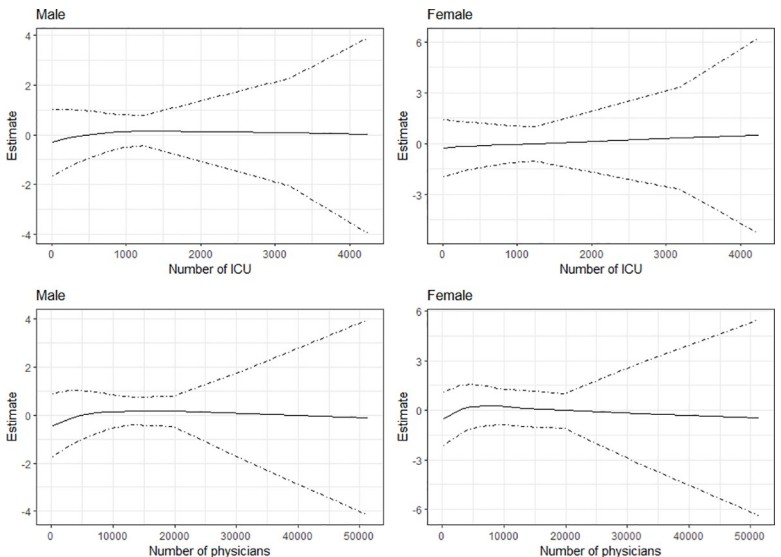

**Fig 3. Estimated effects of number of ICUs and number of physicians on mortality from COVID-19 in the municipalities of Brazil.**

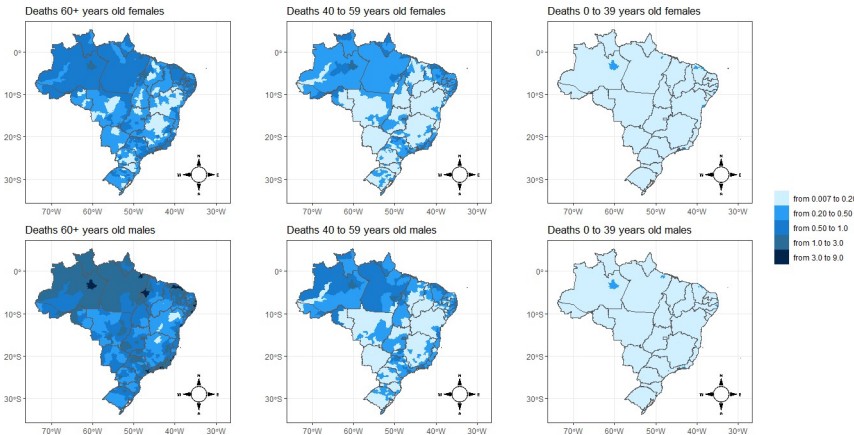

**Fig 4. Spatial pattern of COVID-19 mortality risk by sex and microregion of Brazil, 2020.** Source: Brazilian Ministry of Health 2020.

increase. This is because only few municipalities have large numbers of ICUs and physicians. As evident from the plots, the number of ICUs appears to have little or no effects on infection for both sexes, as the estimates are about zero. For number of physicians, the results show slight rise in infections as the number of physicians becomes greater than 3500, especially for males. In the case of mortality (Fig 3), the posterior mean estimates are about zero for both variables, indicating minimal effects on mortality from COVID-19 in the municipalities of Brazil.

Figs 4 and 5 show the spatial distribution of the posterior means for relative risks for mortality and infection from COVID-19, respectively. The findings from the spatial analysis are not reported in the conventional 5-year age groups, but are instead in three large age groups (0–39, 40–59, and >60 years). We adopt these age ranges because our estimates show that the likelihood of death is quite small for individuals under 40 years of age, but increases slowly up to age 60 and then rises exponentially. Additionally, the age pattern of the relative risks of infection follows a structure different from those of mortality, peaking between the ages of 20–54 years old, a result similar to the findings of another study [13] (data not shown). To use the same scale of comparison for deaths and infections, we opted to present all spatial analyses using these age groups.

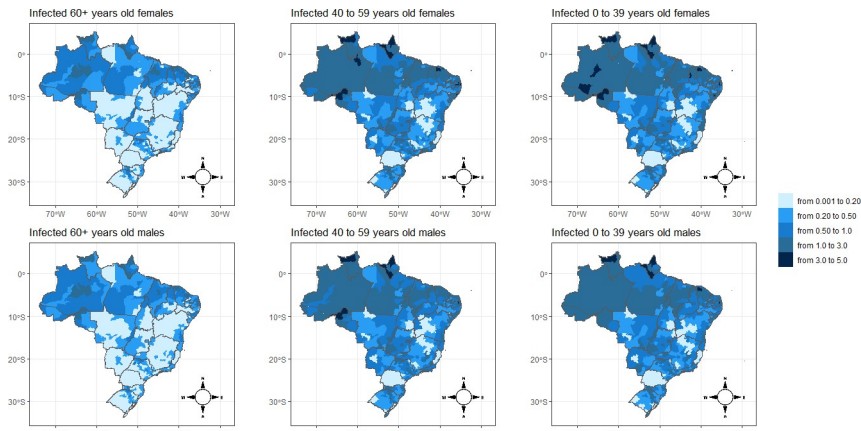

**Fig 5. Spatial pattern of COVID-19 infection risk by sex and microregion of Brazil, 2020.** Source: Brazilian Ministry of Health 2020.

In general, the results show that the relative risks for mortality (Fig 4) by microregion and sex are remarkably higher for older adults. Similar spatial patterns are observed for males and females. There are few distinctions in the geographical distribution of the risks of deaths between males and females across the age groups. From children to young adults, the relative risks almost do not surpass the value of 0.20 across the microregions. Between the ages of 40 and 59, there were isolated spots of high mortality risk, but in few places around the North Amazon Metropolitan area (Manaus) and in the Northeast coast microregion of Fortaleza (Ceará). These two areas were heavily affected in terms of cases and deaths in the first months of the pandemic [7,25].

Additionally, in the Rio de Janeiro and São Paulo microregions, located in the Southeast of Brazil, there are spatial clusters of high mortality risks among this age group. This could be expected because, being business and touristic areas, these two locations were the first to report cases of SARS-CoV-2 infection in the country [7,25]. Once more, it is important that we take into consideration the time that the pandemic hit a certain location, which may affect its relative risks in comparison with other locations. However, this fact does not entirely compromise our estimates because we also see many other areas in the country that registered much later cases of COVID-19 and they are showing relative mortality risks similar to those of the microregions that present more consolidated data.

For the population above 60 years of age, the results indicate additional spatial clusters. There exists a cluster of higher mortality risks that extends from the Southeast part of the country and connects to the Center-West of the country. In addition, North Amazon and the coast of the Northeast regions present a high risk of mortality for the elderly due to COVID-19. One interesting fact is that the highest mortality risks are basically located in Northern and Northeastern coast areas of Brazil, despite the South and Southeast microregions presenting older population age structures [26], and one would expect that in these last regions overall mortality due to COVID-19 might be higher, but this is not the case, especially in the aged South region of the country. We may speculate that socioeconomic disparities and limited access to health facilities in Northern parts of Brazil play a role in increasing mortality risks among elderly groups in these areas.

Using PNS data from 2013, Borges and Crespo [27] show that age is the principal risk factor for comorbidities associated with COVID-19 in Brazil, but sociodemographic variables also have important impacts, generally indicating higher risk for persons in more vulnerable categories, such as those with less schooling and the black and brown population. Their study also shows the prevalence of certain diseases such as diabetes and high blood pressure across different age groups and other SES categories.

A second important finding is that the spatial pattern of infection risks is also highly concentrated in the North and Northeast of Brazil (Fig 5). Here, we see a significant contribution of the young adult population to increasing the infection risk in these areas, while in other parts of the country the risk of COVID-19 infection is much lower. In addition to the socioeconomic dimension, the high infection risk in young adult population could explain the higher risk of mortality in these areas as well.

We now turn our analysis to the decomposition effects. Fig 6 shows the decomposition of the effects of mortality and infection risks in terms of differences in age compositions and other effects, captured by differences in risks. To facilitate the analyses, we grouped the microregions according to macroregions and selected the region with the highest proportion in the oldest age group (the South), as a benchmark of comparison with the others. This way, we were able to analyze how the age structure affects COVID-19 mortality and infections across the country.

Population age structure across regions in Brazil is very different. The population is younger in the North and Northeast and older in the South and Southeast. In 2018, about 10% of

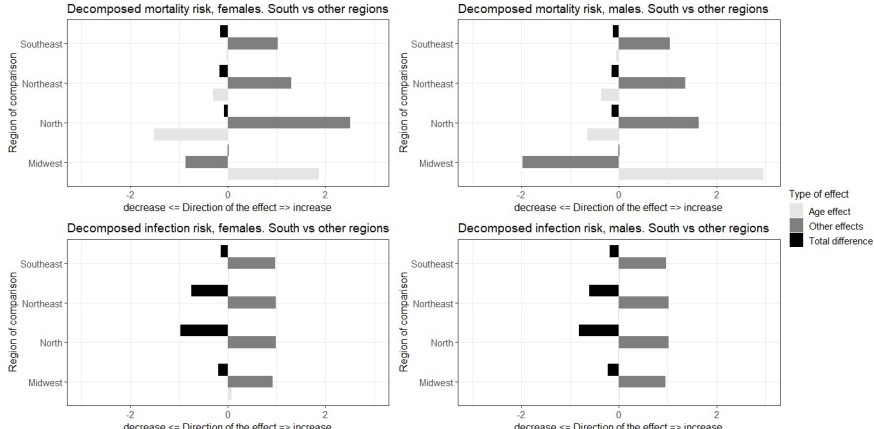

**Fig 6. Differences in mortality and infection risks decomposed by age composition and other effects, comparison between macroregions of Brazil, South versus other regions.** Source: Brazilian Ministry of Health 2020.

the Brazilian population was over 65 years of age, but the variation between Federation Units can range from 13% in Rio Grande do Sul to 7% in Mato Grosso. We cannot directly compare mortality levels from one state and another because it can show the impact of both mortality levels and age structure.

The decomposition exercise did not show major differences among females and males, so we describe the findings for both sexes together. In terms of mortality risks, with the exception of the Center-West, the South region has a lower risk of mortality than the other regions. Age composition tends to reduce these differences, especially relative to the younger populations in the Northern and Northeastern regions, but other elements are responsible for increasing the differences in mortality risk between the South and other parts of Brazil.

When we looked at infection risks, age composition played a small role in the differences between macroregions. All areas presented a higher risk of COVID-19 infection than the South, and many unobserved factors are responsible for these regional differences in the number of cases. This last finding does not exclude the fact that age structure also played an important role in the spatial pattern of infection, as we saw in the previous estimates disaggregated by microregions.

The population is younger in many microregions of the North and Northeast, but in these regions, the highest overall mortality rates are presented, when we control for population age structure. We also found that these areas, even before the pandemic, presented greater risks of general mortality and did not have good health infrastructure [9,28].

## Discussion

In most countries, we do not have only one COVID-19 outbreak going on. There are several differences across regions in the rhythm and stage of the outbreaks [29]. Thus, it is very relevant to investigate the patterns in infections and mortality across different regions of a socioeconomically diverse country like Brazil [30,31]. One important issue is the limitation regarding COVID-19 data in the country due to the lack of adequate testing and reporting. In this study, we use a Bayesian hierarchical model to estimate rates of infection and mortality from COVID-19 based on the microregions of Brazil, taking into account the population age structure and sex distribution, and controlling for the number of ICUs and physicians available to the municipalities as well as the levels of socioeconomic inequality commonly expressed in regional diversity.

Overall, our results suggest a different spatial pattern between the risks of death and the relative risks of infection for both sexes. While the relative mortality risks by microregion and sex are remarkably higher for older adults, particularly for men, the relative risks for infections are concentrated among young adults, being proportionally higher among women. However, mortality risks are much greater for males than females.

The number of ICUs and physicians appear to have only minimal impact on infection and death from COVID-19 in the municipalities of Brazil. A possible explanation for these unexpected findings could be that, in most cases, the number of these facilities is proportional to the population of the municipalities, such that crowded municipalities would have a higher number of ICUs and physicians. Noronha et al. [19] discuss the distance from each city to a main health care hub and show that regions in the North and Northeast have the longest distance to a city with an adequate offer of ICU beds. Also, Pereira et al. [32] show that there is a large variation in access to health care in Brazil. The access to ICUs varies across cities and regions and is much lower in poorer areas.

We find that almost all the regions in the North and a good number of the microregions in the Northeast coast appear to be at a disadvantage in terms of the overall risk of mortality, when compared to the Southeast and South regions of the country. These are important points to be considered when coping with COVID-19, a disease that is more severe in older populations. Even with a younger demographic, many locations in the North and Northeast have higher overall mortality rate, which could be due to worse health conditions overall. In addition, these areas also present a great number of infections. In spite of the protective factor of a more youthful population, the microregions of the North and Northeast of the country have higher mortality rates due to age for certain causes of death that can accentuate the risks of complications from COVID-19 [33,34].

For instance, Baptista and Queiroz [35] observed a rapid increase in mortality from cardiovascular diseases between 2001 and 2015 in the Northern and Northeast parts of the country, whereas in the South and Southeast they found a steady decline. Overall, they also show that in approximately 75% of Brazilian microregions, men have higher mortality rates than women. The North and Northeast regions are the least developed socioeconomically regions of the country and have mortality rates higher than the others. Previous studies show that less developed areas have seen the fastest rate of increase in mortality from cardiovascular diseases in recent years. Finally, diabetes mellitus affects mainly women. In approximately 78% of Brazilian micro-regions, mortality rates are higher for women than men. However, the spatial distribution suggests that, for males, this cause is more heterogeneous when analyzing the country as a whole. Spatial analysis also shows that, for both men and women, there is a concentration of high rates of mortality from diabetes in regions of the North and Northeast of the country.

The Northern and Northeastern regions are the least socioeconomically developed regions of the country and have higher mortality rates than the others. França et al. [34] also show that mortality rates and prevalence of diabetes are concentrated in this region of the country [26,36]. The population is younger in many microregions of the North and Northeast, but that is where the highest overall mortality rates are presented when we control for population age structure. We also found that these areas, even before the pandemic, presented greater risks of general mortality and did not have good health infrastructure [9,28].

This scenario, combined with the weakening of measures to deal with the pandemic, such as the measures indicated by the WHO, are elements that need to be considered in mitigating possible future short- and medium-term regional impacts of this pandemic. There are certain areas that, even before the pandemic, already presented greater risks of general mortality, due to lack of good healthcare infrastructure, a large share of the population that needs emergency assistance from the government, and elderly people with comorbidities who are not practicing

social isolation, even at the moment of the greatest spread of the pandemic. Populations in these areas are exposed to greater risks of collapse in the healthcare system and can give rise to a considerable increase in the number of deaths from COVID-19 [9].

## Author Contributions

**Conceptualization:** Everton Emanuel Campos de Lima, Ezra Gayawan, Emerson Augusto Baptista, Bernardo Lanza Queiroz.

**Data curation:** Everton Emanuel Campos de Lima, Ezra Gayawan, Emerson Augusto Baptista.

**Formal analysis:** Everton Emanuel Campos de Lima, Ezra Gayawan, Emerson Augusto Baptista, Bernardo Lanza Queiroz.

**Investigation:** Everton Emanuel Campos de Lima, Ezra Gayawan, Emerson Augusto Baptista, Bernardo Lanza Queiroz.

**Methodology:** Everton Emanuel Campos de Lima, Ezra Gayawan.

**Supervision:** Everton Emanuel Campos de Lima.

**Visualization:** Everton Emanuel Campos de Lima, Emerson Augusto Baptista, Bernardo Lanza Queiroz.

**Writing – original draft:** Everton Emanuel Campos de Lima, Ezra Gayawan, Emerson Augusto Baptista, Bernardo Lanza Queiroz.

**Writing – review & editing:** Everton Emanuel Campos de Lima, Ezra Gayawan, Emerson Augusto Baptista, Bernardo Lanza Queiroz.

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
