## [Decision Letter · Decision Letter 0]

23 Oct 2020

PONE-D-20-27148

Spatial pattern of COVID-19 deaths and infections in small areas of Brazil

PLOS ONE

Dear Dr. Baptista,

Thank you for submitting your manuscript to PLOS ONE. After careful consideration, we feel that it has merit but does not fully meet PLOS ONE’s publication criteria as it currently stands. Therefore, we invite you to submit a revised version of the manuscript that addresses the points raised during the review process.

Please, make sure you address all points raised by the three reviewers in length, since two of them (reviewers two and three) raised relevant questions about the soundness of the study approach. Also, note that reviewer one made several language edits directly to the submitted document as well as comments and suggestions within comment balloons to the side of the edited text. Make sure you don't miss these comments in the revised paper. Since the methodological questions raised by reviewers two and three are related to the adequacy of the study design, proper addressment of those issues will be deeply taken into account in the final decision of a revised version of the manuscript.

We look forward to receiving your revised manuscript.

Kind regards,

Albert Schriefer, M.D., Ph.D.

Academic Editor

PLOS ONE

Journal Requirements:

2.We suggest you thoroughly copyedit your manuscript for language usage, spelling, and grammar. If you do not know anyone who can help you do this, you may wish to consider employing a professional scientific editing service.  

3. Please note that according to our submission guidelines (http://journals.plos.org/plosone/s/submission-guidelines), outmoded terms and potentially stigmatizing labels should be changed to more current, acceptable terminology. For example: “access backwardness” should be changed to “access challenges” and so forth (as appropriate).

4.Thank you for stating the following in the Acknowledgments Section of your manuscript:

[Gayawan contribution was funded by the São Paulo Research Foundation (FAPESP) [grant

268 2018/18649-7] and Lima and Gayawan thank the University of Campinas Research Fund -

269 FAEPEX for support.]

 [The author(s) received no specific funding for this work00]

5.We note that [Figure(s) 2 and 3] in your submission contain [map/satellite] images which may be copyrighted. All PLOS content is published under the Creative Commons Attribution License (CC BY 4.0), which means that the manuscript, images, and Supporting Information files will be freely available online, and any third party is permitted to access, download, copy, distribute, and use these materials in any way, even commercially, with proper attribution. For these reasons, we cannot publish previously copyrighted maps or satellite images created using proprietary data, such as Google software (Google Maps, Street View, and Earth). For more information, see our copyright guidelines: http://journals.plos.org/plosone/s/licenses-and-copyright.

1.    You may seek permission from the original copyright holder of Figure(s) [2 and 3] to publish the content specifically under the CC BY 4.0 license. 

Reviewers' comments:

Reviewer's Responses to Questions

**Comments to the Author**

1. Is the manuscript technically sound, and do the data support the conclusions?

Reviewer #1: Yes

Reviewer #2: Partly

Reviewer #3: No

2. Has the statistical analysis been performed appropriately and rigorously? 

Reviewer #1: Yes

Reviewer #2: No

Reviewer #3: Yes

3. Have the authors made all data underlying the findings in their manuscript fully available?

Reviewer #1: Yes

Reviewer #2: Yes

Reviewer #3: Yes

4. Is the manuscript presented in an intelligible fashion and written in standard English?

Reviewer #1: Yes

Reviewer #2: Yes

Reviewer #3: No

5. Review Comments to the Author

Reviewer #1: The paper describes the geographic distribution of COVID-19 throughout Brazil, attributing non-homogeneity to demographics specifically age. Specific comments and edits to conform to standard North American English are included in the attached document.

Reviewer #2: The manuscript presents interesting findings. But it needs to include essential confounding variables in the spatial analysis. Regarding mortality, the essential variable in this spatial analysis is the access to intensive care beds. This may explain part of the higher mortality in the north and northeast, which has the lowest number of beds per 100k of the country. It is important in this spatial analysis to incorporate this variable, perhaps it is more important than the distribution by sex and age in these regions. Another variable that can impact mortality is the coverage of primary care and coverage of supplementary private health. In addition, the analysis has to adjust the 558 558 micro-regions for the moment of the first case and count the same first days for the different regions. Only in this way is it possible to verify regional differences in terms of mortality and infection risks.

Reviewer #3: I recommend (i) to make (indirect) standardization by agegroup in the different region to support the analysis comparing the microregions among the Brazilian regions, apparently they may be so different (example: North (Amazon) Region with SouthWest region). (ii) The primary sample unit used did not match with the terciary (i.e ITU bed) health care service provided in macro or microregion defined by Brazilian Health System (SUS) (iii) The authors did not adjust by co-morbidities, however the Brazilian MoH conducted national surveys about non communicable diseases, with data available to make adjust about these diseases; (iv) In other hand, there is available data regarding National health care survey (PNAS) about health services provided by microregion (or cities)...the analysis conducted did not support the conclusions.

6. PLOS authors have the option to publish the peer review history of their article (what does this mean?). If published, this will include your full peer review and any attached files.

Reviewer #1: No

Reviewer #2: **Yes: **Julio Croda

Reviewer #3: No

---

## [Author Response · Author response to Decision Letter 0]

6 Dec 2020

Dear Editor,

A point-by-point response to the reviewers is attached.

Thank you,

Authors

---

## [Decision Letter · Decision Letter 1]

27 Jan 2021

Spatial pattern of COVID-19 deaths and infections in small areas of Brazil

PONE-D-20-27148R1

Dear Dr. Baptista,

We’re pleased to inform you that your manuscript has been judged scientifically suitable for publication and will be formally accepted for publication once it meets all outstanding technical requirements.

Kind regards,

Albert Schriefer, M.D., Ph.D.

Academic Editor

PLOS ONE

Additional Editor Comments (optional):

Reviewers' comments:

Reviewer's Responses to Questions

**Comments to the Author**

1. If the authors have adequately addressed your comments raised in a previous round of review and you feel that this manuscript is now acceptable for publication, you may indicate that here to bypass the “Comments to the Author” section, enter your conflict of interest statement in the “Confidential to Editor” section, and submit your "Accept" recommendation.

Reviewer #3: All comments have been addressed

2. Is the manuscript technically sound, and do the data support the conclusions?

Reviewer #3: Yes

3. Has the statistical analysis been performed appropriately and rigorously? 

Reviewer #3: Yes

4. Have the authors made all data underlying the findings in their manuscript fully available?

Reviewer #3: Yes

5. Is the manuscript presented in an intelligible fashion and written in standard English?

Reviewer #3: Yes

6. Review Comments to the Author

Reviewer #3: The majority of the comments were included in the manuscript. I believe that this manuscript will help to improve the discussion about an uncommon level related to Unified Health System in the Brazil.

7. PLOS authors have the option to publish the peer review history of their article (what does this mean?). If published, this will include your full peer review and any attached files.

Reviewer #3: **Yes: **WILDO NAVEGANTES DE ARAUJO

---

## [Editor Report · Acceptance letter]

1 Feb 2021

PONE-D-20-27148R1 

Spatial pattern of COVID-19 deaths and infections in small areas of Brazil 

Dear Dr. Baptista:

I'm pleased to inform you that your manuscript has been deemed suitable for publication in PLOS ONE. Congratulations! Your manuscript is now with our production department. 

Kind regards, 

on behalf of

Dr. Albert Schriefer 

Academic Editor

PLOS ONE